# Resolving puzzles of the phase-transformation-based mechanism of the strong deep-focus earthquake

Valery I. Levitas [1,2,3] ✉

Deep-focus earthquakes that occur at 350–660 km are assumed to be caused by olivine → spinel phase transformation (PT). However, there are many existing puzzles: (a) What are the mechanisms for jump from geological $10^{-17} - 10^{-15}\,\text{s}^{-1}$ to seismic $10 - 10^3\,\text{s}^{-1}$ strain rates? Is it possible without PT? (b) How does metastable olivine, which does not completely transform to spinel for over a million years, suddenly transform during seconds? (c) How to connect shear-dominated seismic signals with volume-change-dominated PT strain? Here, we introduce a combination of several novel concepts that resolve the above puzzles quantitatively. We treat the transformation in olivine like plastic strain-induced (instead of pressure/stress-induced) and find an analytical 3D solution for coupled deformation-transformation-heating in a shear band. This solution predicts conditions for severe (singular) transformation-induced plasticity (TRIP) and self-blown-up deformation-transformation-heating process due to positive thermomechanochemical feedback between TRIP and strain-induced transformation. This process leads to temperature in a band, above which the self-blown-up shear-heating process in the shear band occurs after finishing the PT. Our findings change the main concepts in studying the initiation of the deep-focus earthquakes and PTs during plastic flow in geophysics in general.

Deep-focus earthquakes are very old puzzles in geophysics. While the shallow earthquakes occur due to brittle fracture, materials at 350–600 km are under pressure of 12–23 GPa and temperature of 900–2000 K and are above the brittle-ductile transition[1]. That is why the main hypothesis is that the earthquakes are caused by instability due to phase transformation (PT) from the subducted metastable $\alpha$-olivine ($\text{Mg}_x\,\text{Fe}_{1-x})_2\text{SiO}_4$ to denser $\beta$-spinel or $\gamma$-spinel[2–11] (Fig. 1a); for the San Carlos olivine $x = 0.9$. Self-organized ellipsoidal transformed regions (anticracks) filled with nanograined product phase with very low shear resistance and orthogonal to the largest normal stress were considered. A set of anticracks aligned along the maximum shear stress reduces shear resistance and causes a shear band. In refs. 12, 13, the acoustic emission approach was pioneered to detect "seismic" events during several PTs, which was interpreted in favor of PT and shear instability hypotheses of the earthquake initiation. The modern acoustic emission approach combined with microstructural analyses is presented in refs. 10, 14, 15. However, we will show that these semi-qualitative approaches cannot resolve the existing puzzles. In particular, the mechanisms for jumping from geological $10^{-17} - 10^{-15}$ to seismic $10 - 10^3\,\text{s}^{-1}$ strain rates (see[4]) are not understood, and it is not clear whether they are possible without PT. Next, abrupt olivine-spinel PT in seconds, while it does not occur for over a million years, needs to be quantitatively rationalized. Deviatoric strain-dominated seismic signals caused by volume-change-dominated transformation strain[1,9] should also follow from some equations.

[1]Iowa State University, Department of Aerospace Engineering, Ames, IA 50011, USA. [2]Iowa State University, Department of Mechanical Engineering, Ames, IA 50011, USA. [3]Ames Laboratory, Division of Materials Science and Engineering, Ames, IA 50011, USA. ✉e-mail: vlevitas@iastate.edu

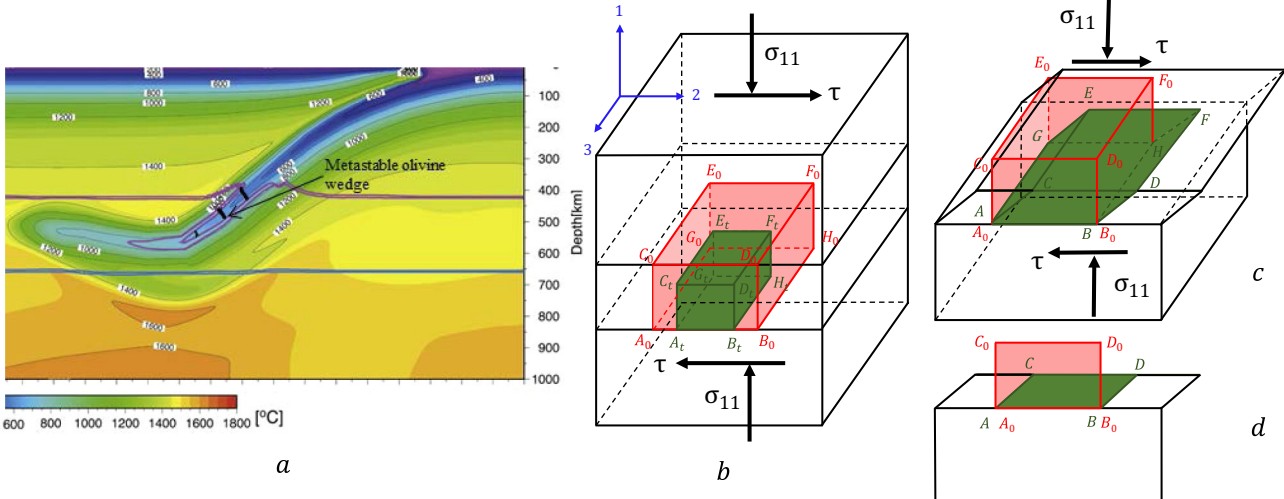

**Fig. 1 | Schematics of triggering deep-focus earthquake by transformation-deformation-heating bands during phase transformation (PT) from the subducted metastable olivine to spinel. a** Results of modeling of subduction of the Pacific plate including metastable olivine wedge beneath Japan with the temperature contour line. Magenta lines denote 1% (upper line) and 99% (lower line) of PT from olivine to $\beta$-spinel; blue lines denote 1% (upper line) and 99% (lower line) of PT from $\gamma$-spinel to bridgmanite+magnesiowüstite. Black lines designate transformation-deformation-heating bands. Earthquakes occur at the olivine wedge boundary (adapted with modifications from ref. 11 with permission from Elsevier Publ.). **b** Schematics of a transformation-deformation-heating band within a rigid space. Part of a band before PT (red) and after PT and isotropic transformation strain (green) is shown. **c** To satisfy the continuity of displacements across the shear-band boundary and rigid space outside the band, additional transformation-induced plasticity (TRIP) develops, leading to deformation of the green rectangular $A_tB_tG_tH_t$ to $ABGH$ that coincides with $A_0B_0G_0H_0$ and to large plastic shear. **d** 2D view (along axis 3) of **c**.

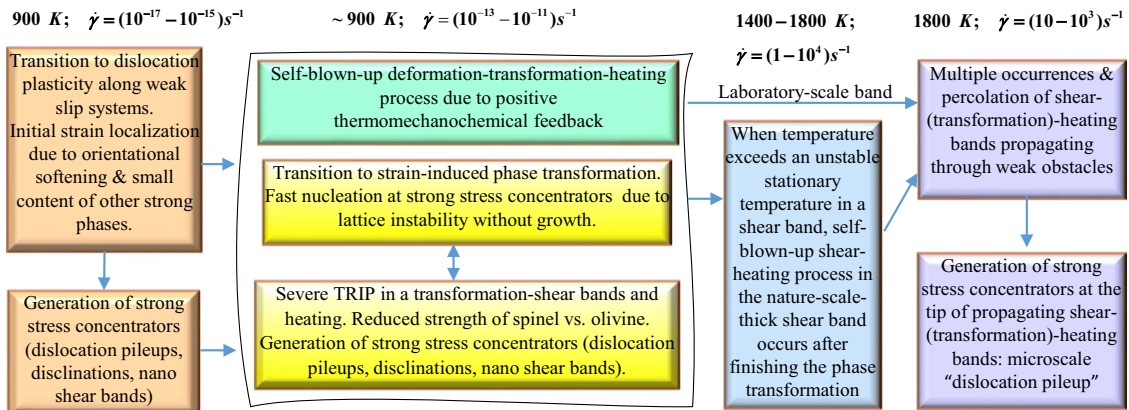

**Fig. 2 | Mechanisms of localized thermoplastic flow and phase transformation (PT) leading to high strain and PT rates and high temperatures in a transformation-deformation and shear bands.** Temperature and shear rate before each stage is shown on the top. Initial strain localization occurs due to transition to dislocation plasticity along properly oriented weak [001](010) slip systems and corresponding orientational softening, as well as along the path with a small content of other strong phases like diopside. Localized plastic flow leads to the generation of strong stress concentrators (dislocation pileups, disclinations, shear nanobands), causing strain-induced PT. Due to crystal lattice instability, fast nucleation at strong stress concentrators occurs during 10 ps but without growth, leading to a weaker nanograined spinel and strain-controlled kinetics proportional to the strain rate instead of time. Volume reduction during PT in a shear band causes severe transformation-induced plasticity (TRIP), which in turn causes strain-induced PT leading to further TRIP and PT, and so on. This positive thermomechanochemical feedback leads to self-blown-up deformation-transformation-heating up to high temperature, exceeding unstable stationary temperature in a shear band, and high-strain rate. After completing PT, in nature but not in the thin laboratory-scale band, further heating and increased strain rate in a band occur due to shear flow. Similar processes are expected in multiple transformation-deformation bands and then just deformation bands that find ways through weak obstacles and may percolate or just increase the total shear-band volume and amplify generated seismic waves. Propagating transformation-deformation and just plastic shear bands generate strong stress concentrators at their tips producing a microscale counterpart of a dislocation pileup, which causes both fast PT and plasticity and further propagation of a shear band, i.e., repeats the above processes at a larger scale.

In this work, we suggested mechanisms of localized thermoplastic flow and PT that consist of several interrelated steps shown in Fig. 2. We introduce a combination of several novel concepts that allow us to resolve the above puzzles quantitatively. We treat the olivine-spinel PT as plastic strain-induced (instead of pressure/stress-induced), which was not done for any PT in geophysics. This leads to completely different kinetics, for which the transformation rate is proportional to the strain rate, explaining very high transformation rate for very high-strain rates. We find an analytical 3D solution for TRIP and coupled PT-TRIP-heating processes in a shear band. This solution predicts conditions for severe (singular) TRIP and self-blown-up deformation-PT-heating process due to positive thermomechanochemical feedback between TRIP and strain-induced transformation, leading to completing the PT in a few seconds. Severe TRIP shear explains shear-dominated seismic signals. In nature, this process leads to temperature in a band exceeding the unstable stationary temperature, above which

the self-blown-up shear-heating process in the shear band continues after completing the PT. Without PT and TRIP, significant temperature and strain rate increase is impossible. Due to the much smaller shear band thickness in the laboratory, there is no heating, and plastic flow after the PT is very limited. Our results change the main concepts in studying the deep-focus earthquakes and PTs during plastic flow in geophysics in general.

## Results and discussion

### Utilizing high-pressure mechanochemistry

It is clear that to obtain such jumps in plastic flow and PT rates in some rare cases, a theory should contain singularity that strongly depends on some external conditions. To resolve the problem, we will utilize the main concept of high-pressure mechanochemistry[16–19]. Our first point is that in all previous geophysical papers[2–10,20], pressure- and stress-induced PTs were considered a mechanism for initiating the shear instability. These PTs start at crystal defects naturally existing in material and for stresses below the yield strength. These defects (e.g., various dislocation structures or grain boundaries) produce stress concentrators and serve as nucleation sites for a PT. Since the number of such defects is limited, one has to increase pressure to activate defects with smaller stress concentrations. In contrast, plastic strain-induced PTs occur by nucleation at defects produced during plastic flow. The largest concentration of all stress components can be produced at the tip of the dislocation pileups, proportional to the number of dislocations $N$ in a pileup. Since $N = 10 - 100$, local stresses could be huge and exceed the lattice instability limit, leading to the nucleation of spinel within sub-nanoseconds, which is negligible compared to the $1 - 10$ s time scale considered here. Indeed, a typical time for the loss of lattice stability and reaching a new stable phase for different PTs obtained with molecular dynamics simulation is <10 ps[21–24]. Due to a strong reduction of stresses away from the defect tip, growth is very limited. Thus, the next plastic strain increment leading to new defects and new nuclei at their tips is required to continue PT. That is why (and because of barrierless nucleation, which does not require thermal fluctuations) time is not a governing parameter in a kinetic equation, and plastic strain plays a role of a time-like parameter[16–19,25] (Eq. 4). Arrested growth also explains nanograin structure after strain-induced PTs in various systems[25–28], including olivine → spinel[4,6,10,29]. The important point is that the deviatoric (nonhydrostatic) stresses in the nanoregion near the defect tip are not bounded by the engineering yield strength but rather by the ideal strength in shear for a defect-free lattice which may be higher by a factor of 10–100. Local stresses of such magnitude may result in the nucleation of the high-pressure phase at an applied pressure that is not only significantly lower than that under hydrostatic loading but also below the phase-equilibrium pressure. For example, plastic strain-induced PT from graphite to hexagonal and cubic diamonds at room temperature was obtained at 0.4 and 0.7 GPa, 50 and 100 times below than under hydrostatic loading, respectively, and well below the phase-equilibrium pressure of 2.45 GPa[26] (see other examples for PTs in Zr, Si, and BN[25,27,30,31]). In addition, such highly-deviatoric stress states with large stress magnitudes cannot be realized in bulk. Such unique stresses may lead to PTs into stable or metastable phases that were not or could not be attained in bulk under hydrostatic or quasi-hydrostatic conditions[25,27,32,33]. It was concluded in refs. 16–19 that plastic strain-induced transformations require completely different thermodynamic, kinetic, and experimental treatments than pressure- and stress-induced transformations.

Thus, our quantitative mechanisms of very fast localized thermoplastic flow and PT consist of several interrelated steps shown in Fig. 2 and contain several conceptually important points:

(a) Proof that plastic flow alone cannot lead to localized in mm-scale band heating, that is why PT is required.

(b) Substitution of stress-induced PT with plastic strain-induced PT, which was not previously used in geophysics and leads to completely different kinetic description. Transformation rate is proportional to the strain rate, which explains very high transformation rate for very high-strain rates.

(c) Transition to dislocation flow with strong stress concentrators is required to substitute stress-induced PT with barrierless and fast plastic strain-induced PT.

(d) Strain-induced PT in a shear band generates severe (singular) TRIP shear and heating, which in turn produces strain-induced PT and so on, resulting in the self-blown-up PT-TRIP-heating process due to positive thermomechanochemical feedback. This process leads to completing the PT on the few second time scale. Severe TRIP shear explains shear-dominated seismic signals.

(e) The self-blown-up PT-TRIP leads to the heating above the unstable stationary temperature $T_s = 1400 - 1800$ K, after which further heating in a shear band occurs due to traditional thermoplastic flow. Achieving $T = 1800$ K is sufficient to reach $\dot{\varepsilon}(T) = 10 - 10^3$ s$^{-1}$ and generate strong seismic waves.

(f) These processes repeat themselves at larger scale.

Lack of any of these processes due to not meeting the required conditions (e.g., proper orientation or path with a small content of stronger phases) may lead to inability to reach very fast localized PT and plastic flow and cause an earthquake, which explains why the strong earthquakes are relatively rare events. Similarly, lack of seismic activity below 660 km, where endothermic and slow disproportionation reaction from γ-spinel to bridgmanite+oxide (magnesiowüstite) occurs, can be explained.

Relatively small shear strain in laboratory experiment[29] ($\gamma = 43$ vs. $\gamma = 10^6$ in nature) is because the temperature cannot grow due to an extremely thin band; processes in the third column in Fig. 2 are absent, and TRIP occurs only. Our Eq. 1 below relates the change in strain rate with respect to the initial one before localization. That is why the final strain rate is distributed with depth similar to the initial strain rate before localization. This is consistent with the correlation between seismicity in the transition zone and strain rate before localization[34].

### Mechanisms and conditions of localized thermoplastic flow and heating in Mg$_{1.8}$Fe$_{0.2}$SiO$_4$ olivine

According to[34], seismicity in the transition zone correlates with the rate of plastic flow, which is in the range of $10^{-17} - 10^{-15}$ s$^{-1}$. Orthorhombic olivine has only three independent slip systems set, i.e., less than five required for the accommodation of arbitrary homogeneous deformation. That is why other mechanisms like grain-boundary migration through disclination motion[35], amorphization[36], dislocation climb, diffusive creep, and other isotropic mechanisms with linear flow rule[37,38] supplement dislocation plasticity and control strain rate. Less than 40% of olivine aggregate strain at high temperatures may be accommodated by dislocation activity. However, when one of the slip systems is aligned along or close to maximum shear stress, faster shear-dominated deformation is possible controlled solely by dislocations. Especially, [001](010) slip system has critical shear stress of 0.15 MPa, at least three times lower than that for all other systems (at 405 km depth, $T = 1757$ K, $p = 13.3$ GPa, equivalent plastic strain rate $\dot{\varepsilon} = 10^{-15}$)[38]. Thus, if some group of grains is oriented with [001](010) slip system along the maximum shear stress, dislocation glide may occur compatible with shear strain localization due to orientational softening. Despite the variety of deformation mechanisms, plastic flow in olivine is formally described by

$$\dot{\varepsilon} = H\sigma^n \exp(-Q_r/T) \rightarrow M = \dot{\varepsilon}(T)/\dot{\varepsilon}(T_0) = \exp[-Q_r(T^{-1} - T_0^{-1})], \quad (1)$$

where $Q_r = Q/R$, $Q$ is the activation energy, $R$ is the gas constant, and $\sigma$ is the differential stress, which is approximately the same within and outside of the shear band due to continuity of shear and normal stresses along the band boundary. Since for olivine $n = 3.5$[38,39], reduction in slip resistance by a factor of 3 leads for the same stress to increase in the strain rate by a factor of 47. Also, in Earth, olivine is mixed with other phases, e.g., diopside, which has much higher critical shear stresses, 7.31-64.7 MPa and $n = 6.4 - 11.4$ at the same conditions[38], and which may constitute 30% of the olivine-diopside mixture. Thus, shear localization should start in the region with small diopside content, bypassing diopside inclusions, which may also increase strain rate by additional two–three orders of magnitude. In total, when both proper alignment of olivine grains and small diopside content are combined, the local strain rate may increase at least by $10^4$ times without a change in temperature and reach $10^{-13} - 10^{-11}$ s$^{-1}$. At such a strain rate, shear localization may be promoted by plastic heating in a band with the width $h$ exceeding $10 - 10^3$ m[39], but a characteristic time of this localization, $10 - 10^4$ years, is way too long to resolve puzzles mentioned in abstract, and too broad to reproduce a few-mm thick slip zone in the Punchbowl Fault[4,6]. Also, such a slow heating increases chances for slow and nonlocalized olivine-spinel PT, which eliminates the possibility of fast and localized PT and TRIP described in the next section.

To estimate softening due to the substitution of olivine to a weaker nanograined spinel in a band, we will use data from ref. 40. The initial yield strength in compression $\sigma_y$ of the transformed nanograined $\gamma$-spinel at $\dot{\varepsilon} \simeq 10^{-5}$ s$^{-1}$ is 4.7 times lower than that for olivine. The estimated strain rate in Earth in this nanograined $\gamma$-spinel is $10^{-13}$ s$^{-1}$. This shows, in contrast to ref. 4, 6, that weak nanograined spinel cannot even close provide the seismic strain rate $10 - 10^3$ s$^{-1}$. Note that the strength completely recovers within 5 h due to grain growth. Anticracks filled with weaker nanograined spinel along the path of a shear band also reduce strength (the main softening mechanism suggested in refs. 2, 4, 6), but much less than the above estimate when nanograined spinel is located within the entire shear band; that is why we will not consider them. While we included reduced strength of spinel versus olivine in Fig. 2, we did not use it in our estimates, getting more conservative values.

We assume that the initial temperature of the cold slab is $T_0 = 900$ K[40], cold enough to avoid stress-induced olivine-spinel PT in bulk, and show that to get the desired jump in the strain rate, the final temperature should be $T = 1800$ K. Indeed, taking from ref. 39 $Q_r = 58,333$ K we obtain from Eq. 1 that at $T = 1800$ K the strain rate increases by a factor of $M = 10^{14}$ (Fig. 3a). Thus, if initial strain rate in the localized region was $\dot{\varepsilon}(T_0) = 10^{-13} - 10^{-11}$ s$^{-1}$, then after heating to $T = 1800$ K it increases to $\dot{\varepsilon}(T) = 10 - 10^3$ s$^{-1}$. While we did not include

spinel in our calculations, these numbers are close to strain rates of $1 - 10$ s$^{-1}$ for $\gamma$-spinel obtained for San Carlos olivine at 17 GPa, 1800 K, and grain size of $10 nm$ that can be estimated from Fig. S10 in ref. 40. Thus, despite the doubt of the validity of Eq. 1 for such high-strain rates, it gives a reasonable order-of-magnitude value.

The temperature evolution equation in a localized shear band with the thickness $h$ and temperature $T$ within the rest of the material with temperature $T_0$ is

$$\rho v \dot{T} h = -4k(T - T_0)/h + \sigma_y \dot{\varepsilon} h = -4k(T - T_0)/h + H\sigma^{n+1}\exp(-Q_r/T)h, \quad (2)$$

where $\rho$ is the mass density, $v$ is the specific heat, and $k$ is the thermal conductivity. The term $-4k(T - T_0)/h$ is the heat flux through two shear-band surfaces due to temperature gradient $2(T - T_0)/h$, similar to ref. 39, and Eq. 1 was used to calculate plastic dissipation. The thermal conductivity $k = \rho v \kappa = 2.4 \times 10^{-6}$ MPa m$^2$/(s K)[39], where $\kappa = 10^{-6}$ m$^2$/s is the thermal diffusivity, $\rho = 3000$ kg/m$^3$, and $v = 800$ J/(kg K) $= 800 \times 10^{-6}$ MPa m$^3$/(kg K). Constant $H$ is determined from Eq. 1 as $H = \dot{\varepsilon}(T_0)\sigma^{-n}\exp[Q_r/T_0]$. Then the stationary solution $T_s$ of Eq. 2 (i.e., $\dot{T} = 0$) is determined from

$$T_s - T_0 = 0.25h^2\sigma\dot{\varepsilon}(T_0)\exp[-Q_r(T_s^{-1} - T_0^{-1})]/k. \quad (3)$$

Since the Punchbowl Fault exhibited a few-mm thick slip zone[4,6], we assume $h = 4 \times 10^{-3}$ m. We also choose $\sigma = 300$ MPa[39,40].

Plots of both sides of Eq. 3 in Fig. 3b shows that there are two stationary solutions. One of the solutions with $T \simeq T_0$ is stable since any fluctuational increase (decrease) in temperature within a band leads to higher (lower) heat flux from the band than the plastic dissipation. The second solution $T_s \gg T_0$ varies from 1396 to 1825 K when strain rate $\dot{\varepsilon}(T_0)$ reduces from $10^{-10}$ to $10^{-14}$ s$^{-1}$. The higher combination $h^2\sigma\dot{\varepsilon}(T_0)$ is, the lower the stationary temperature $T_s$ is. This solution is unstable since any fluctuational increase (decrease) in temperature within a band leads to higher (lower) plastic dissipation than the heat flux from the band and further increase (decrease) in temperature. This means that (a) localized increase in strain rate and temperature in a thin band is impossible, and temperature increase estimated with neglected heat flux term to justify melting[41] or low shear resistance[4,6] are wrong; (b) some very significant additional heating source than the traditional plastic flow is required to reach $T_s$; otherwise, the temperature will be close to $T_0$; (c) after reaching $T_s \gg T_0$, plastic dissipation will lead to unlimited heating up to melting temperature with a corresponding drastic increase in the strain rate. Thus, even if the entire olivine would transform everywhere to much weaker nanograined spinel (not just in selected anticracks) and softening due to small content of other strong

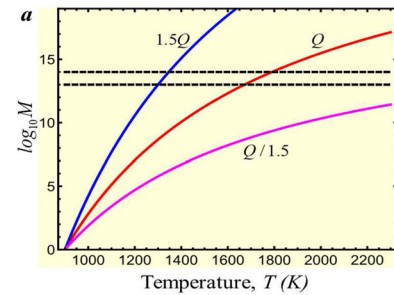
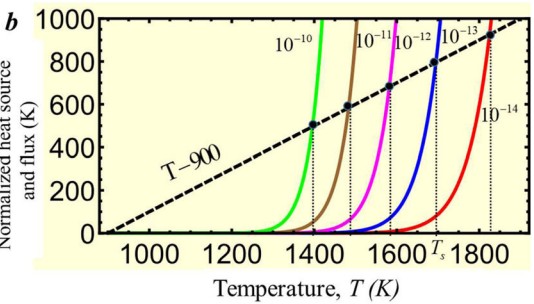
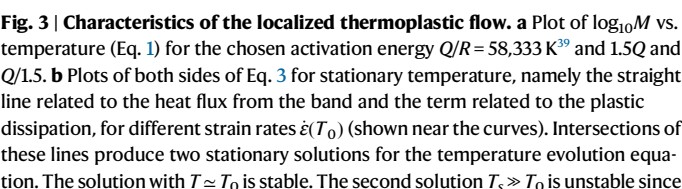

**Fig. 3 | Characteristics of the localized thermoplastic flow. a** Plot of $\log_{10}M$ vs. temperature (Eq. 1) for the chosen activation energy $Q/R = 58,333$ K[39] and $1.5Q$ and $Q/1.5$. **b** Plots of both sides of Eq. 3 for stationary temperature, namely the straight line related to the heat flux from the band and the term related to the plastic dissipation, for different strain rates $\dot{\varepsilon}(T_0)$ (shown near the curves). Intersections of these lines produce two stationary solutions for the temperature evolution equation. The solution with $T \simeq T_0$ is stable. The second solution $T_s \gg T_0$ is unstable since

any fluctuational increase (decrease) in temperature within a band leads to higher (lower) plastic dissipation than the heat flux from the band and further increase (decrease) in temperature. This means that (i) some very significant additional heating source than the traditional plastic flow is required to reach $T_s$; otherwise, the temperature will be close to $T_0$; (ii) after reaching $T_s$, plastic dissipation will lead to unlimited heating up to melting temperature with a corresponding drastic increase in the strain rate.

phases (which were not included in the previous models) are present, still, strain rate cannot exceed $\dot{\varepsilon}(T_0) = 10^{-13} - 10^{-11}$ s$^{-1}$, which cannot cause a localized temperature increase.

Note that the transformation heat for olivine-spinel PT increases temperature by 100 K only[42], which is too small to reach $T_s \gg T_0$. Below, we suggest PT- and TRIP-related mechanisms of increase in temperature above $T_s$.

## Plastic strain-induced phase transformation olivine → spinel

Usually, during a PT, spinel appears as a continuous film along grain boundaries with increasing thickness[43] or as anticrack region nucleated at the grain boundaries[4,6,29]. Transition to dislocation plasticity should lead to dislocation pileups and strain-induced PT within grains, consistent with band-shaped spinel regions observed within grains in refs. 4, 6, 29 and related to dislocation pileups. It is known that large overdrive and nonhydrostatic stresses promote martensitic PT at dislocations within grains[44,45]. Shear stresses at the tip of the dislocation pileup should also change a slow reconstructive mechanism of olivine-spinel PT to a fast martensitic mechanism; however, this is not a necessary condition for our scenario. Transformation bands include (010) planes, which include [001](010) slip system with the smallest critical stress, see ref. 38, consistent with our assumption above. However, there are also (011) transformation bands, which do not have smaller critical shear stress and do not lead to orientational softening. That means that orientational softening is not a mandatory mechanism for initial localization and can be compensated by smaller diopside content along those planes.

Strain-controlled kinetic equation[17,18] for the volume fraction of the strain-induced high-pressure phase simplified in Supplementary Information is

$$\frac{dc}{d\varepsilon} = A(1-c) \quad \text{for } p > p_\varepsilon^d(T); \quad A := a\frac{p - p_\varepsilon^d(T)}{p_h^d(T) - p_\varepsilon^d(T)} \quad \rightarrow \quad c = 1 - \exp(-A\varepsilon).$$

(4)

Here, $p_\varepsilon^d(T)$ and $p_h^d(T)$ are the minimum pressure at which the direct (i.e., to high-pressure phase) strain-induced and pressure-induced PTs are possible, respectively, and $a$ is a parameter. We do not consider strain-induced reverse spinel → olivine PT because the resultant nanograin spinel deforms dominantly by grain-boundary sliding, which does not produce stress concentrators inside the grains. The first experimental and only existing confirmation of Eq. 4 and parameter identification were performed for $\alpha \rightarrow \omega$ PT in Zr[31]. Based on data, $A \simeq 23$ for $p = p_e$, which will be used due to lack of data for olivine → spinel PT. While we study the effect of $A$ on the transformation kinetics (Fig. 4), it is shown below that for large shear strains, the term with $A$ is negligible in the expression for TRIP.

In contrast to pressure/stress-induced PT, time is not a parameter in Eq. 4; plastic strain plays a role of a time-like parameter. Thus, the rate of strain-induced PT is determined by the rate of plastic deformation. To reach $c = 0.99$, plastic strain $\varepsilon = 4.6/A = 0.2$, which at strain rate 10 s$^{-1}$ (or, alternatively, at 10$^{-4}$ s$^{-1}$) takes just 0.02 s (or, alternatively, 20 s), instead of millions years without plastic strain. Thus, plastic strain can increase the transformation rate by >12–16 orders of magnitude.

Strain-induced character of PTs is consistent with results in refs. 4, 6, 10, 29, where metastable olivine Mg$_2$GeO$_4$ (structural analog of natural olivine that transforms at much lower pressure) transforms into spinel in the 70 nm thick shear band, partially transforms in the surrounding band of few $\mu m$ thick, and does not transform away from the band. These thin planar layers of strain-induced nanograined (10–30 nm) Mg$_2$GeO$_4$ spinel within olivine were observed in refs. 4, 10, 29 after laboratory experiment and suggested as an additional to anticrack mechanism of shear weakening. They appear along the specific slip planes, are related to dislocation pileups, and correspond to our model's prediction below. The lower temperature is, the more strain-induced planar spinel bands and less stress-induced spinel anticrack regions are observed, consistent with promoting effect of strain-induced defects. Relative slip along a 70 nm thick transformed planar layer is 3 microns, i.e., shear strain $\gamma = 43$; slip rate is 1 $\mu$m s$^{-1}$, thus $\dot{\gamma} = 14$ s$^{-1}$ and time of sliding (and PT) is $\gamma/\dot{\gamma} = 3$ s[4,29]. These bands offset multiple non-transforming pyroxene crystals, which allows for determining relative slip. In contrast to anticracks that are mostly orthogonal to the compressive stress, transformation bands are mostly under 45$^0$ with some scatter to the compression direction, i.e., they coincide with planes with maximum shear stress or pressure-dependent resolved shear stress.

Similar results are obtained for silicate olivine Fe$_2$SiO$_4$[15] tested at pressure range 3.9 – 8.4 GPa and temperature range 748 – 923 K. Co-seismic slip of 40 microns over the fault width of 1.5 microns, i.e., an order of magnitude larger than in germanium olivine, results in $\gamma = 27$, i.e., the same order of magnitude as in germanium olivine. While faults in Mg$_2$SiO$_4$ and Mg$_{1.8}$Fe$_{0.2}$SiO$_4$ have not been observed yet, due to the close magnitude of the transformation strain for all (Mg$_x$ Fe$_{1-x}$)$_2$SiO$_4$ and (Mg$_x$ Fe$_{1-x}$)$_2$GeO$_4$ for any $x$ (see supplementary materials), similar $\gamma$ is expected.

In nature, the Punchbowl Fault also exhibited a few-mm thick slip and PT zone, along which slip occurred by several kilometers, which contains product nanograins[4,6], i.e., shear strain $\gamma = 10^6$. Similar strain-induced PTs and reactions are observed at the surface layers in friction experiments[4,6].

## TRIP and self-blown-up deformation-transformation-heating process

Next, we need to find a mechanism for a drastic increase in strain rate and temperature. We suggest that TRIP caused by olivine → spinel PT can lead to this. TRIP occurs due to internal stresses caused by volume change during the PT combined with external stresses. We found (Supplementary Information) an analytical 3D solution, in which the plastic shear $\gamma$, which is TRIP, is related to the applied shear stress $\tau$, the yield strength in shear $\tau_y$ during PT, and volumetric transformation strain $\varepsilon_o$ (see Fig. 4a) as

$$d\gamma/dc = \frac{2}{\sqrt{3}}|\varepsilon_o|(\tau/\tau_y)/\sqrt{1 - \left(\tau/\tau_y\right)^2} \rightarrow \gamma = \frac{2}{\sqrt{3}}c|\varepsilon_o|(\tau/\tau_y)/\sqrt{1 - \left(\tau/\tau_y\right)^2}.$$

(5)

Effective transformation volumetric strain $c\varepsilon_o$ during growth of $c$ forces plastic strain to restore displacement continuity across an interface (see Fig. 1b, c), and plastic flow takes place at arbitrary (even infinitesimal) shear stress. The yield strength in shear $\tau_y$ during PT is unknown. Atomistic simulations for many materials (e.g., in refs. 26, 46) show that lattice resistance drops to and even below zero after lattice instability. For strain-induced PT, nanosize nuclei also reduce the yield strength[40]. We assume conservatively that $\tau_y = const = \sigma/\sqrt{3} = 173$ MPa. For $\tau \rightarrow \tau_y$ (e.g., in a shear band), plastic shear tends to infinity (Fig. 4a). This is the desired singularity we wanted to find above. Note that our 3D solution has the proportionality factor $2\sqrt{3} \simeq 3.4$ times larger than in the previous 2D treatments[47–50], which changes the current results qualitatively.

Since PT causes TRIP, which (like traditional plasticity) promotes strain-induced PT, it, in turn, promotes TRIP, and so on, there is positive thermomechanochemical feedback, which we called a self-blown-up deformation-transformation-heating process. In such a case, Eq. 4 cannot be integrated alone but should be considered together with Eq. 5. For shear-dominated flow $\varepsilon = \gamma/\sqrt{3}$, and we obtain

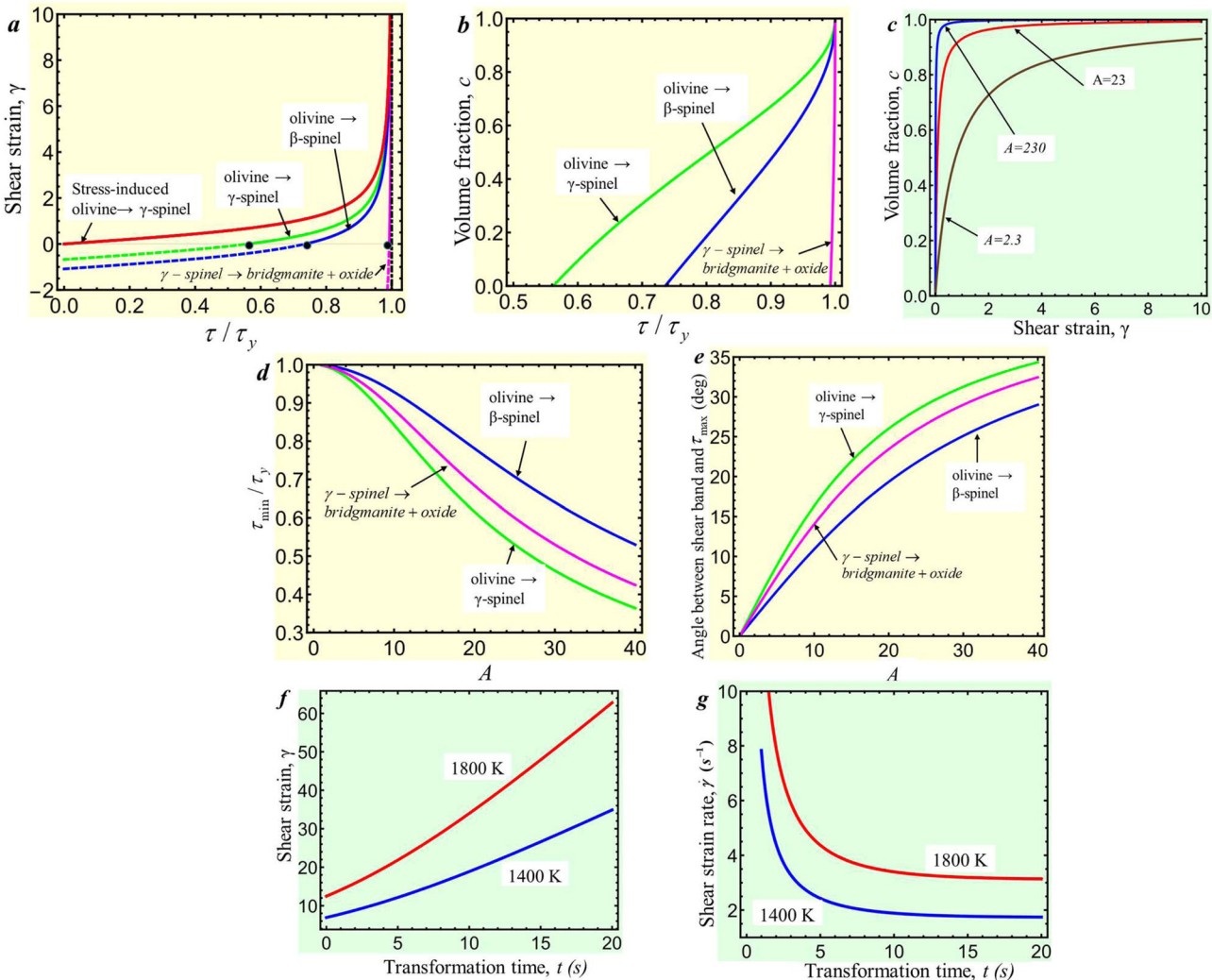

**Fig. 4 | Kinetics of coupled strain-induced phase transformations and transformation-induced plasticity (TRIP). a**, **b** Shear strain and volume fraction of the high-pressure phase vs. $\tau/\tau_y$, respectively. Dots denote shear stress $\tau_{min}$ for initiation of strain-induced phase transformation (PT). Line for the stress-induced PT corresponds to Eq. 5. **c** Kinetics of olivine → $\gamma$-spinel PT for different kinetic parameters $A$. **d**, **e** Shear stress $\tau_{min}/\tau_y$ for initiation of strain-induced PT and angle $\alpha$ between the shear-transformation band and direction with maximum shear stress $\tau_{max}$, respectively, vs. kinetic parameter $A$. Results for chemical reaction $\gamma$-spinel → bridgmanite+oxide (magnesiowüstite) are included in **a**–**e** with $A = 2.3$ and $\varepsilon_0 = 0.08^3$. **f**, **g** Shear strain and shear strain rate, respectively, required to reach temperatures of 1400 K and 1800 K during transformation time $t$ for parameters for the Punchbowl Fault.

(Fig. 4a–d)

$$\gamma = 2\frac{|\varepsilon_o|}{\sqrt{3}}\frac{\tau}{\tau_y}\bigg/\sqrt{1-\left(\frac{\tau}{\tau_y}\right)^2}-\sqrt{3}/A; \quad c = 1-\frac{3}{2}\sqrt{1-\left(\frac{\tau}{\tau_y}\right)^2}\bigg/\left(\frac{\tau}{\tau_y}A|\varepsilon_o|\right)$$
$$= \left(1+\sqrt{3}/(A\gamma)\right)^{-1}; \tag{6}$$

$$\tau/\tau_y \geq 1/\sqrt{1+4A^2|\varepsilon_o|^2/9}. \tag{7}$$

Equation 7 is the criterion for a self-blown-up deformation-transformation-heating process, shown in Fig. 4d vs. $A$. It is obtained from Eq. 6 and condition $c \geq 0$ or $\gamma \geq 0$. The last expression for $c(\gamma)$ in Eq. 6 is obtained by excluding $\tau/\tau_y$ from two previous Eqs. 6. For $Mg_{1.8}Fe_{0.2}SiO_4$ olivine → $\gamma$-spinel PT $\varepsilon_o = -0.096$ and for olivine → $\beta$-spinel PT $\varepsilon_o = -0.06$, see refs. 3, 51 and supplementary material; this results in $\tau/\tau_y \geq 0.562$ for $\gamma$-spinel and $\tau/\tau_y \geq 0.736$ for $\beta$-spinel, which are not very restrictive. Thus, since $\tau/\tau_y = \cos 2\alpha$, where $\alpha$ is the angle between maximum shear stress and shear band, the above criterion is met at $\alpha \leq 27.9^o$ for $\gamma$-spinel and

$\alpha \leq 21.3^o$ for $\beta$-spinel (Fig. 4e). We will focus on olivine → $\gamma$-spinel PT since it has larger TRIP and less restrictive constraints.

To have $\gamma = 10$, $\tau/\tau_y = 0.999939$ and $c = 0.9925$; for $\gamma = 100$, $\tau/\tau_y = 0.999999$ and $c = 0.999248$. Thus, for the self-blown-up deformation-transformation process to produce shear $\gamma > 10$, one needs $\tau/\tau_y = 1$, i.e., perfect alignment of maximum shear stress and shear band. This contributes to understanding why the self-blown-up deformation-transformation-heating process and strong deep-focus earthquakes are relatively rare processes. Equation 7 explains extremely large shear strains (sliding) in a fault or friction surface. Also, since the shear strain is much $> \varepsilon_o$, this resolves a puzzle of the shear character of the deep-earthquake source[1,9]. Note that for very large TRIP shear the term $-\sqrt{3}/A$ in Eq. $6_2$ is negligible (Fig. 4a), i.e., TRIP shear is independent of any kinetic properties (specifically, parameter $A$) of strain-induced PT. Also, for $\tau/\tau_y \to 1$, Eq. $6_2$ gives $c \to 1$. TRIP-induced temperature rise is determined by the equation

$$\rho \nu \dot{T} h = -4k(T-T_0)/h + \tau_y \dot{\gamma} h, \tag{8}$$

in which for $\tau \to \tau_y$ we even neglected the transformation heat to have a conservative estimate. The solution is

$$T = T_0 + (T_s^{tr} - T_0)\left[1 - \exp\left(-\frac{4k}{\rho\nu h^2}t\right)\right]; \quad T_s^{tr} = T_0 + \frac{\tau_y \dot{\gamma} h^2}{4k}, \quad (9)$$

where $T_s^{tr}$ is the stationary temperature due to TRIP heating. The shear rate to reach temperature $T$ during the PT time $t$, as well as corresponding shear strain $\gamma$ are determined from Eq. 9

$$\dot{\gamma} = (T - T_0)\frac{4k}{\tau_y h^2}\left[1 - \exp\left(-\frac{4k}{\rho\nu h^2}t\right)\right]^{-1}; \quad \gamma = \dot{\gamma}t; \quad \gamma(t=0) = \frac{\rho\nu}{\tau_y}(T - T_0). \quad (10)$$

Note that $M$ in Eq. 1, $T_s$ in Eq. 3, and Eqs. 8–10 are independent of the exponent $n$ in Eq. 1. Figure 4f, g exhibit $\dot{\gamma}$ and $\gamma$ required to reach temperatures 1800 K and 1400 K vs. transformation time $t$ for parameters for the Punchbowl Fault. The faster PT is, the smaller shear but larger strain rates are required. Minimum shears are at $t = 0$ (instantaneous PT), $\gamma(1800) = 12.5$ and $\gamma(1400) = 6.9$ but lead to infinite strain rate. For $t < 10$ s, the desired temperature is reached during transitional heating. For $t > 10$ s, it is reached by approaching a stationary temperature; that is why the required strain rates approach stationary values. Based on kinetic estimates in ref. 40, time for complete pressure-induced PT at 17 GPa and 1420 K is 10 s; strain-induced PT may occur by orders of magnitude faster even at a much lower temperature.

Practically, limitation comes from the required shear (rather than the shear rate). For $t \leq 10$ s, the required strain is <43 observed in the laboratory[4,29]. Based on Eq. 6, strain $\gamma \geq 10$ requires $\tau/\tau_y \geq 0.999939$, i.e., practically perfect alignment of the shear band along the maximum shear direction. The shear rate is calculated by dividing shear by PT time. For $t > 1$ s, shear rate is $s < 10$ s$^{-1}$, and after completing PT it further increases during traditional plastic flow due to $T > T_s$ (Fig. 3b). For $0.001 < t < 1$ s, the shear rate is in the range of $10 - 10^4$ s$^{-1}$, on the same order of magnitude as expected at 1800 K during traditional plastic flow.

Thus, TRIP and the self-blown-up deformation-transformation-heating process should lead to temperatures >$T_s$ in Fig. 3, after which further drastic temperature increase does not need PT and can occur due to traditional plastic flow. Note that since during PT $\tau/\tau_y \simeq 1$, traditional plastic flow (which is neglected) should add to TRIP and further increase both strain rate and temperature.

Theoretically, thermoplastic unstable temperature increase above $T_s$ can lead to melting, which is one of the mechanisms of high-strain rate shear localization and deep earthquake[1,41]. However, due to a strong heterogeneity of earth materials along the shear band, including non-transforming minerals, melting temperature (which is around 2700 K at 17 GPa for $Mg_2SiO_4$ and $Mg_{1.8}Fe_{0.2}SiO_4$[52]) may not be reached and is not necessary. As estimated above, reaching 1800 K is sufficient for achieving strain rates $10 - 10^3$ s$^{-1}$. We also want to stress that the melting-based mechanism of the deep earthquake is possible in nature only if some other processes (like self-blown-up deformation-transformation-heating) will increase temperature above $T_s$.

Similar processes are expected in multiple transformation-shear and shear bands (Fig. 2) that find ways through weak obstacles and may percolate or just increase the total shear-band volume and amplify generated seismic waves. In reality, the shear band is not infinite but has a very large (10 to 1000 and larger) ratio of length, at least in the shear direction, to the width. That is why the above theory is applicable away from the tips of a band. When finite-size single or coalesced deformation or transformation-deformation bands propagate, stresses at their ends are equivalent to those at a dislocation pileup or superdislocation, but at a larger scale[53] and with the total Burgers

vector $\gamma h$, which may be huge. These stresses cause both fast PT and plasticity and further propagation of shear band and trigger initiation of new bands, mostly mutually parallel. Such a stress concentrator is by a factor of $\gamma/\varepsilon_0$, i.e., orders of magnitude, stronger than that at the tip of the anticrack[2–6,8,29] and much more effective in spreading transformation-deformation bands at the higher, microscale. The resulting propagating thermoplastic band can pass through non-transforming minerals and extend outside the metastable olivine wedge. Indeed, it was demonstrated in ref. 6 that the fault originated in metastable $Mg_2GeO_4$ olivine during its transformation to spinel propagated through previously transformed spinel.

## Analysis of the lack of seismic activity below 660 km

Lack of any of the processes shown in Fig. 2 due to not meeting the required conditions may explain lack of seismic activity below 660 km, where endothermic and slow disproportionation reaction from ring-woodite to $MgSiO_3$ (bridgmanite) + $(Mg_x Fe_{1-x})O$ (magnesiowüstite) occurs. It is difficult to say which exactly process is missing because a counterargument may override each argument. For example, one may say that the chemical reaction, in contrast to the martensitic PT, requires a diffusive mass transport, and both nucleation and growth cannot be as fast as martensitic PT, which is proved for the proxy reaction albite → jadeite + coesite[6,54]. However, this may be true or not because large plastic shears strongly accelerate mass transport and chemical reactions as well[49,55–59], and it is unknown how do shears affect this specific reaction. In particular, at friction surfaces the decomposition reaction of dolomite $MgCa(CO_3)_2 \to MgO + CaO + 2CO_2$ completes within 0.006 s[4] with temperature increase exceeding 1000 K. That is why the martensitic character of PT is not required here and was not required for olivine → spinel PT because reconstructive PT can also be drastically accelerated by plastic straining.

The most probable reasons are:

(a) lack of initial shear localization in nanograined spinel before reaction due to grain sliding deformation without orientational softening (which reduces $\varepsilon(T_0)$ by a factor of 47) and reduced dislocation activity, which makes the transition to strain-induced PT and self-blown-up deformation-transformation-heating process impossible;

(b) the higher initial temperature at 660 km (see refs. 11, 34 and Fig. 1a); e.g., increase in $T_0$ from 900 K to 1000 K reduces parameter $M$ in Eq. 1 by a factor of 653, and

(c) low initial strain rate below 660 km[34] reduces the final strain rate proportionally.

One of the conditions for PT-induced instability mentioned in refs. 3,6 is the exothermic character of the olivine-spinel PT, leading to runaway heating. At the same time, the reaction from ringwoodite to bridgmanite+magnesiowüstite is endothermic and cannot produce instability and earthquakes below 600 km. However, for coupled strain-induced PT-TRIP process, plastic heating occurs during PT, and the contribution of PT heat (100 K[42]) in temperature increase from 900 to $T_s = 1400 - 1800$ K is small. Thus, we do not think that the exothermic character of PT alone is critical. In laboratory experiments, temperature change within the shear band is negligible.

Exothermic PT was utilized in ref. 4 also to explain nanograined spinel structure. The temperature increase due to PT heat increases the driving force for PT and causes runaway nucleation under growth-inhibited conditions. Suppose a slight temperature increase would be the reason for a drastic increase in nucleation rate. In that case, runaway nucleation should occur everywhere rather than to localize within anticracks, especially in hotter regions of the metastable olivine slab closer to its boundary with spinel. It is also unclear why growth is slow at such a large thermodynamic driving force that causes runaway nucleation. At the same time, nucleation at dislocations and

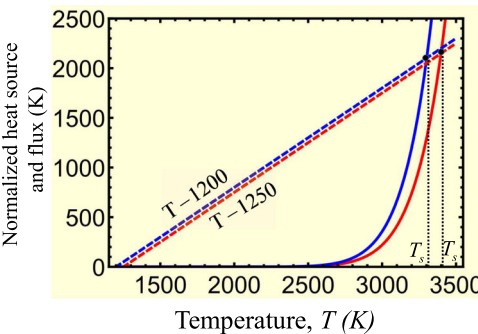

**Fig. 5 | Analysis of experiments in refs. 4,29.** Plots of both sides of Eq. 3 for stationary temperature, namely the straight line related to the heat flux from the band and the term associated with the plastic dissipation, for two different sets of experiments in refs. 4,29. Blue lines correspond to the experiment at $T_0 = 1200$ K and red lines are for $T_0 = 1250$ K. Since unstable stationary temperatures $T_s$ for both experiments are very high, they cannot be reached by thermoplastic flow alone, and a phase transformation with transformation-induced plasticity is required.

dislocations pileups leads to nanograined structure because of growth arrest due to a strong reduction of stresses away from the defect tip[16–19].

## Heat transfer analysis of laboratory experiments[4,29]

Substituting in Eq. 3 data for $Mg_2GeO_4$ from ref. 29, namely (sample GL707), $\dot{\varepsilon}_0 = 2 \times 10^{-4}$ s$^{-1}$, $T_0 = 1250$ K, $\sigma = 1589$ MPa, and $h = 10^{-7}$ m, as well as from ref. 4, $\dot{\varepsilon}_0 = 10^{-4}$ s$^{-1}$, $T_0 = 1200$ K, $\sigma = 1804$ MPa, and $h = 0.7 \times 10^{-7}$ m, we obtain $T_s = 3398$ K for the first case and $T_s = 3302$ K for the second case (Fig. 5). Due to very small shear band thickness in the laboratory experiments, these values are extremely high, far away from the region of stability of spinel, and well above the melting temperature. Since no traces of reverse PT to olivine and melting were observed in refs. 4, 29, these temperatures were not reached, and no thermoplastic shear localization is possible without PT, TRIP, and self-blown-up deformation-transformation-heating process.

However, even with TRIP, substituting in Eq. 9 data from the same laboratory experiment[4] $h = 0.7 \times 10^{-7}$ m, $\dot{\gamma} = 14$ s$^{-1}$, and maximum $\tau_y = 300$ MPa from Fig. S2 in ref. 4, we obtain that the maximum (stationary) temperature increase is just $1.3 \times 10^{-6}$ K. This should not be surprising because thickness $h = 70$ nm in a laboratory experiment is smaller than in Earth $h = 4$ mm by a factor of 57143. Since stationary temperature increment is proportional to $h^2$, for $h = 4$ mm, $\dot{\gamma} = 14$ s$^{-1}$, and $\tau_y = 300$ MPa, it would be $4.33 \times 10^3$ K. Thus, in laboratory experiments on $Mg_2GeO_4$[4] temperature increase in the transformation-shear band was absent.

In ref. 4, adiabatic approximation was used to estimate maximum shear stress and internal friction coefficient from the condition that temperature increment does not exceed 230 K, maximum increment to reach the olivine-spinel phase-equilibrium temperature. A paradoxical result was that the estimated shear stress and friction coefficient were an order of magnitude lower than directly measured. The reason for this paradox is in adiabatic approximation; when heat flux from the shear band is included, the temperature increase is negligible for any reasonable shear resistance and does not restrict the internal friction stress. As it was found in ref. 40, the initial yield strength in compression $\sigma_y$ of the transformed nanograined $\gamma$-spinel at $\dot{\varepsilon} \simeq 10^{-5}$ s$^{-1}$ is 4.7 times lower than that for olivine. The above result also means that the sliding should drastically increase after completing PT; that is why shear in the Punchbowl Fault, $\gamma = 10^6$, is drastically larger than in the laboratory, $\gamma = 43$. Consequently, processes in the third column in Fig. 2 are absent in laboratory experiments and cannot be verified due to small shear band thickness.

Similarly, drastic heating leading to melting and dissociation is predicted in ref. 41 using adiabatic approximation. When heat flux is included, conditions for melting are quite restrictive.

## Relation to some previous works

TRIP is well known to the geological community, but it was considered to have a small effect[7,44,60,61]. This is correct in general, but for a properly oriented shear band where $\tau \to \tau_y$, plastic shear tends to infinity (Eq. 7 and Fig. 4a). Shear banding and TRIP are observed in DAC experiments in fullerene[62] and BN[28] despite the PTs to stronger high-pressure phases. For PT from hexagonal to superhard wurtzitic BN, TRIP was evaluated to be 20 times larger than the prescribed shear[28]. Shear banding during PT is possible if the yield strength $\tau_y$ during PT does not increase despite the high-strain rate and strength of the high-pressure phase, which supports our conservative hypothesis $\tau_y =$ const. Positive feedback between PT and TRIP without heating was suggested in ref. 28 but without any equations. Reaction-induced plasticity (RIP), similar to TRIP, was revealed for a chemical reaction within a shear band in Ti-Si powder mixture[49], and RIP-induced adiabatic heating was considered as a factor promoting reaction rate. However, mechanochemical feedback was not claimed since kinetics was considered within the theory for stress-induced reactions instead of strain-induced.

Here, we follow the main idea formulated in refs. 2–8 that the deep-focus earthquakes can be initiated by instability caused by PT, in particular, from olivine to spinel. However, as we discussed above, the broadly observed self-organized anticracks filled with weak nano-grained spinel aligned along the maximum normal stress direction cannot cause the jump in strain rate by a factor of $10^{18}$. Instead, we use here strain-induced PT in thin planar layers leading to nanograined spinel observed in refs. 4, 10, 29.

It is also demonstrated in the paper that adiabatic approximation for a thin shear band, used to estimate the shear strength in ref. 4 and the possibility of melting in ref. 41, and a corresponding increase in strain rate is wrong. Allowing for the heat flux changes results qualitatively.

It is shown in ref. 63 based on the elegant dynamic solution for "pancake-like" flattened ellipsoidal Eshelby inclusion that it can grow self-similarly above some critical pressure. It is also derived that in order for the total strain energy to be finite (and not zero) in the inclusion with tending to zero thickness, deviatoric eigen strain (without specification of its nature) must tend to infinity (even under hydrostatic compression), which "explains" deviatoric character of the deep-earthquake source. This argument is unphysical: why should zero-thickness inclusion "desire" to have nonzero strain energy? Eigen strain in inclusion should be determined by processes in inclusion, like PT and plasticity, which is done in the current paper. Huge TRIP shear in Eq. 6 after complete PT explains deviatoric character of the deep-earthquake source. Also, plasticity (that significantly affects the stress-strain fields, reduces thermodynamic driving force, and may arrest PT[64]) is neglected in ref. 63, as well as interfacial energy.

Our findings change the main concepts in studying the initiation of the strong deep-focus earthquakes and PTs during plastic flow in geophysics in general. They will be elaborated in much more detail using modern computational multiscale approaches for studying coupled PTs and plasticity[16], which can describe nucleation and evolution of multiple PT-shear bands from nano- to macroscales[53,65,66]. They will also be checked in experiments with rotational diamond anvil cell[26–28,31,33] in a closed feedback loop with simulations. Introducing strain-induced PT and the self-blown-up transformation-TRIP-heating process may change the interpretation of various geological phenomena. In particular, they may explain possibility of the appearance of microdiamond directly in the cold Earth crust within shear bands[26] during tectonic activities without subduction to the high-pressure and high-temperature mantle and uplifting. Developed theory of the self-

blown-up transformation-TRIP-heating process is applicable outside geophysics for various processes in materials under pressure and shear, e.g., for new routes of material synthesis, friction and wear under high load, penetration of the projectiles and meteorites, surface treatment, and severe plastic deformation and mechanochemical technologies[16–19,32,56–59].

## Methods

Analytical methods used in the paper are described in the main text and Supplementary Material.

## Data availability

All data needed to evaluate the conclusions in the paper are present in the paper and/or the Supplementary Materials.

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

## Acknowledgements
Support from NSF (CMMI-1943710 and DMR-1904830) and Iowa State University (Vance Coffman Faculty Chair Professorship) is greatly appreciated.

## Author contributions
V.I.L. is the sole author of the results obtained in the current paper.

## Competing interests
The author declares no competing interests.
