## [Peer Review File · Nature Communications]

Resolving puzzles of the phase-transformation-based mechanism of the deep-focus earthquakeREVIEWER COMMENTS

Reviewer #1 (Remarks to the Author):

This paper uses thermomechanical equations to reveal the quantitative mechanism of a localized thermoplastic flow accompanied with the phase transformation for deep-focus earthquakes. The discrepancy between laboratory results and geophysical observations of deep-focus earthquakes has been debated for many years. Although several previous studies proposed the combination of phase transformation and thermal instability to explain the diversity of characteristics of deep-focus earthquakes, laboratory experiments could not replicate the combination, and the quantitative explanation was lacking. The author focuses on the plastic flow accelerating the phase transformation and shear deformation. I was impressed by the work bridging the temporal-spatial difference between laboratory results and geophysical observations. This will provide important insights into the puzzles for deep-focus earthquakes. The paper is worth publishing in Nature Communications, but I have several comments.

Major comments

1. The author uses constant stress exponent $n=3.5$ derived from dislocation creep in Eqs. (2) and (3). However, nanocrystalline spinel inside shear bands have a possibility to deform by diffusion creep due to its fine grain size (e.g., Karato et al., 2001PEPI; Shimojuku et al., 2009EPSL). In the case of diffusion creep ($n=1$), T_s in Eq. (3) will become much higher. Its T_s can exceed the melting temperature. I would like to suggest that the author refers to this point in the paper.

Minor comments

1. On page 1, the author shows nucleation of spinel during ~ 10 ps when $N=10-100$. How does the author calculate 10 ps? Or please cite references.
2. The definition of plastic flow is a bit confusing. In the paper, the plastic flow follows the dislocation or diffusion creep's flow law, right?
3. The total strain consists of plastic strain and transformational strain (Eq. (S.4)). Is elastic strain ignored?
4. According to Burnley et al. (2019AGU), anticracks were composed of a small number of spinel crystals and, in many cases, composed of only one crystal instead of nanocrystalline spinel.
5. Reference [34] Ogava->Ogawa

Reviewer #2 (Remarks to the Author):

Title: Resolving puzzles of the phase-transformation-based mechanism of the strong deep-focus earthquake

Author: Valery I. Levitas

This paper suggests a possible resolution of a problem related to mechanism of deep-focus earthquakes, in terms of transformation-induced plasticity (TRIP) coupled to strain-induced transformation, which results in "self-blown-up" (in author's terminology) shear heating in a shear band thus promoting the transformation on a millisecond time scale.

The mere idea of using TRIP to explain deep earthquakes is not new, but in this paper it is worked out in great numerical detail. The author has been involved in TRIP for at least 20 years and has generated important result; e.g., structural changes in boron nitride under a combination of pressure and shear (Appl. Phys. Lett. 86, 071912 (2005)). So, there is no doubt that the author is an expert in the field of TRIP and strained-induced transformations. And this work may be suitable for publication in Nature Communications as it addresses issues of extreme importance for physics and geophysics. However, the current version of this work reads confusing and needs a lot of improvements before it can be considered for publication in this journal. In what follows, I express my concerns and list the points I would ask the author to properly address. I will be willing to re-evaluate the revised version.

1. What material does the paper discuss? I can only guess it is Mg_2SiO_4 , by the names of its solid structures (forsterite, wadsleyite, and ringwoodite). Another substance is mentioned in the text, namely, Mg_2GeO_4 . So, what is it: Mg_2SiO_4 or Mg_2GeO_4 , or their mixture? What about Fe_2SiO_4 which is also one of the main constituents of the deep Earth? From my understanding, the material that has to be analyzed is $(\text{Mg-Fe})_2\text{SiO}_4$ as the most realistic case. If it is difficult to repeat an analysis similar to that in this work, at least the author should comment on how his conclusions may change in the realistic case of the Mg-Fe solid solution.

2. Some results throughout the paper that lead to other important results are based on what seems to be arbitrary assumptions. For instance, on p. 4 it is assumed that "... at a higher strain rate increase in stress is slightly higher [than 4.7], 5.8 ..." (by the way, this sentence contains a typo: "in" instead of "is"). Why 5.8? How would choosing a different value (high or lower than 5.8) influence the conclusions of this work?

3. Similarly, in Supplementary Information (SI), while discussing the solid-solid phase boundary under nonzero strain, the author assumes that a transition from olivine to gamma-spinel occurs below 1695 K, and that from olivine to beta-spinel above 1295 K. The author should comment on how good these assumptions are, and how much below/above the transition temperatures may be. Phase diagrams under nonzero strain are not studied well enough, e.g., how different they are from (quasi-)hydrostatic phase diagrams, so this point may be of some concern to many potential readers.

4. How reliable is the assumption " $s=1$ " in Eq. (S.1) which leads to the analytic solution $c=1-\exp(-Ae)$, (S.2), for the volume fraction of the high-P phase? The author operates in terms of strain-controlled kinetic equations. I need to see how they couple to time evolution equations, e.g., in classical nucleation theory, which give $c=1-\exp(-Kt^n)$ for the time evolution of c , where t is time and the rate function K strongly depends on temperature. Coupling both equations may provide another insight on a mechanism for a drastic increase in strain rate and temperature that the author suggests in this work.

5. How may the account of realistic crystal structure (polycrystallinity, heterogeneities like grain boundaries, grain edges, crystal defects, etc.) potentially influence the main findings of this work?

6. Can the deep-focus earthquake mechanism discussed in this paper be coupled to real subduction dynamics to shed more light on some phenomena that remain not fully understood? E.g., relative orientation of the compression (P-) and tension (T-) axes of non-shallow earthquakes in the upper and lower parts of the downgoing slab, or shear failure in the surrounding rocks as a result of implosion related to the olivine-spinel transition, and some related phenomena like the absence of monopolar rarefaction, etc.?

7. There is a number of typos throughout the paper, e.g., "0.02s (or 20s)" on p. 6; I assume the author means 20 ms. I won't list them now. If the author resubmits his work and they (some of them) remain in the revised version which I will get a chance to review, then I will get back to this point.

If this is going to be a Nature publication, everything it presents should be very clear to read and understand, and all the assumptions must be justified.

Finally, the paper would gain more scientific weight if the theoretical work it presents were supplemented by possible experimental study on plasticity in Mg_2SiO_4 under P-T conditions close to those in the deep Earth, or some relevant computer simulations. The author analyzes some experimental data on Mg_2GeO_4 , but again, I believe the actual material discussed in Mg_2SiO_4 or $(\text{Mg-Fe})_2\text{SiO}_4$ solid solution.

Response to the Reviewers' comments

Reviewer #1.

Reviewer's comment:

This paper uses thermomechanical equations to reveal the quantitative mechanism of a localized thermoplastic flow accompanied with the phase transformation for deep-focus earthquakes. The discrepancy between laboratory results and geophysical observations of deep-focus earthquakes has been debated for many years. Although several previous studies proposed the combination of phase transformation and thermal instability to explain the diversity of characteristics of deep-focus earthquakes, laboratory experiments could not replicate the combination, and the quantitative explanation was lacking. The author focuses on the plastic flow accelerating the phase transformation and shear deformation. I was impressed by the work bridging the temporal-spatial difference between laboratory results and geophysical observations. This will provide important insights into the puzzles for deep-focus earthquakes. The paper is worth publishing in Nature Communications, but I have several comments.

Author's response:

I greatly appreciate the Reviewer's positive evaluation of the main results of my paper.

Reviewer's comment:

Major comments

1. The author uses constant stress exponent $n=3.5$ derived from dislocation creep in Eqs. (2) and (3). However, nanocrystalline spinel inside shear bands have a possibility to deform by diffusion creep due to its fine grain size (e.g., Karato et al., 2001PEPI; Shimojuku et al., 2009EPSL). In the case of diffusion creep ($n=1$), T_s in Eq. (3) will become much higher. Its T_s can exceed the melting temperature. I would like to suggest that the author refers to this point in the paper.

Author's response:

First, I added after Eq. (10):

"Note that M in Eq. (1), T_s in Eq. (3), and Eqs. (8)-(10) are independent of the exponent n in Eq. (1)." This already resolves the problem.

In fact, I do not do any calculations with these equations for spinel (since the desired strain rate is already achieved at the end of phase transformation), just for olivine. During olivine-spinel transformation, I assume constant yield strength and n is not involved. Also, after phase transformation in a shear band, strain rates in spinel are $10 - 10^3 s^{-1}$. While data for such high strain rates are not available, based on extrapolations in a more recent Karato et al., 2020 paper [40, Fig. S10], dislocation rather than diffusional creep is most probable.

I understand that the Reviewer's comment may be caused by some wording between Eqs. (1) and (2), which mention spinel. I changed them, utilizing known experimental data instead of Eq. (1). They are not used to prove my hypotheses, just to disprove the anticrack mechanism.

At the same time, I keep in the text the piece which shows consistency of the obtained estimates with those in [40]:

"Thus, if initial strain rate in the localized region was $\dot{\epsilon}(T_0) = 10^{-13} - 10^{-11} s^{-1}$, then after heating to $T = 1800K$ it increases to $\dot{\epsilon}(T) = 10 - 10^3 s^{-1}$. While we did not include spinel in our calculations, these numbers are close to strain rates of $1 - 10 s^{-1}$ for γ -spinel at 17 GPa, 1800 K, and grain size of $10nm$ that can be estimated from Fig. S10 in [40]. Thus, despite the doubt of the validity of Eq.(1) for such high strain rates, it gives a reasonable order-of-magnitude value."

Reviewer's comment:

Minor comments

1. On page 1, the author shows nucleation of spinel during 10 ps when $N=10-100$. How does the author calculate 10 ps? Or please cite references.

Author's response:

I elaborated this statement and included corresponding references:

"Since $N = 10 - 100$, local stresses could be huge and exceed the lattice instability limit, leading to the nucleation of spinel during within sub-nanoseconds, which is negligible compared to the $1 - 10 s$ time

scale considered here. Indeed, a typical time for the loss of lattice stability and reaching a new stable phase for different PTs obtained with molecular dynamics simulation is less than 10 ps [21-24].”

Even if for spinel nucleation time is 10 ms instead of 10 ps, nothing changes, because it is much smaller than 1 – 10 s.

Reviewer’s comment:

2. The definition of plastic flow is a bit confusing. In the paper, the plastic flow follows the dislocation or diffusion creep’s flow law, right?

Author’s response:

It is stated in the first line in the first box in Fig. 2 (and described in the text) that one of the mandatory mechanisms is ”transition to dislocation plasticity along weak slip systems.” I discussed specific slip systems along which orientational softening occurs. The reason for this is that plastic strain-induced phase transformation is caused by dislocation pileup. Our model does not involve diffusion mechanism; it is only discussed in the literature review.

Reviewer’s comment:

3. The total strain consists of plastic strain and transformational strain (Eq. (S.4)). Is elastic strain ignored?

Author’s response:

I added highlighted text in the description near this equation.

”We consider the homogeneous state of the space and band before strain localization and phase transformations (i.e., including elastic strain) as the reference state. Then change in elastic strains during PT is small and can be neglected.”

Reviewer’s comment:

4. According to Burnley et al. (2019AGU), anticracks were composed of a small number of spinel crystals and, in many cases, composed of only one crystal instead of nanocrystalline spinel.

Author’s response:

I included a slightly modified statement:

”The estimated strain rate in Earth in this nanograined γ -spinel is $10^{-13} s^{-1}$. This shows, in contrast to [4,6], that weak nanograined spinel cannot even close provide the seismic strain rate $10 - 10^3 s^{-1}$. Note that the strength completely recovers within 5 hours due to grain growth. Anticracks filled with weaker nanograined spinel along the path of a shear band also reduce strength (the main softening mechanism suggested in [2,4,6]), but much less than the above estimate when nanograined spinel is located within the entire shear band; that is why we will not consider them. While we included reduced strength of spinel versus olivine in Fig. 2, we did not use it in our estimates, getting more conservative values.”

Thus, grains grow in time, probably even until single grain remains. Independently, I do not use anticracks in my models.

In fact, the inability of anticrack to produce earthquakes is supported in an email from Pam Burnley that she wrote to me on 12/15/2021:

”I have gone back to working on this problem in the last few years. We have demonstrated that anti-cracks don’t exist (the features exist but they are not ’anti-cracks’) and are focusing our work on microstructures associated with shear (more along the lines of Kirby’s early work).”

Reviewer’s comment:

5. Reference [34] Ogava→Ogawa

Author’s response: Thank you for the correction.

I greatly appreciate the Reviewer’s time and efforts in examining my paper, useful critical comments, detected misprint, and positive attitude. I took the Reviewer’s comments very seriously and made corresponding changes to the text, improving the paper’s quality.

Reviewer #2 (Remarks to the Author):

Title: Resolving puzzles of the phase-transformation-based mechanism of the strong deep-focus earthquake Author: Valery I. Levitas

This paper suggests a possible resolution of a problem related to mechanism of deep-focus earthquakes, in terms of transformation-induced plasticity (TRIP) coupled to strain-induced transformation, which results in "self-blown-up" (in author's terminology) shear heating in a shear band thus promoting the transformation on a millisecond time scale.

The mere idea of using TRIP to explain deep earthquakes is not new, but in this paper it is worked out in great numerical detail.

Author's response:

I believe that it is not just "great numerical detail," it is changing generally accepted opinion to the opposite. It is mentioned in the paragraph "Relation to some previous works."

"TRIP is well known to the geological community, but it was considered to have a small effect [7,44,60,61]. This is correct in general, but for a properly oriented shear band where $\tau \rightarrow \tau_y$, plastic shear tends to infinity (see Eq.(7) and Fig. 4(a))."

Reviewer's comment:

The author has been involved in TRIP for at least 20 years and has generated important result; e.g., structural changes in boron nitride under a combination of pressure and shear (Appl. Phys. Lett. 86, 071912 (2005)). So, there is no doubt that the author is an expert in the field of TRIP and strained-induced transformations. And this work may be suitable for publication in Nature Communications as it addresses issues of extreme importance for physics and geophysics.

Author's response:

I greatly appreciate the Reviewer's positive evaluation.

Reviewer's comment:

However, the current version of this work reads confusing and needs a lot of improvements before it can be considered for publication in this journal. In what follows, I express my concerns and list the points I would ask the author to properly address. I will be willing to re-evaluate the revised version.

1. What material does the paper discuss? I can only guess it is Mg_2SiO_4 , by the names of its solid structures (forsterite, wadsleyite, and ringwoodite). Another substance is mentioned in the text, namely, Mg_2GeO_4 . So, what is it: Mg_2SiO_4 or Mg_2GeO_4 , or their mixture? What about Fe_2SiO_4 which is also one of the main constituents of the deep Earth? From my understanding, the material that has to be analyzed is $(\text{Mg-Fe})_2\text{SiO}_4$ as the most realistic case. If it is difficult to repeat an analysis similar to that in this work, at least the author should comment on how his conclusions may change in the realistic case of the Mg-Fe solid solution.

Author's response:

I completely agree that I should focus on natural $\text{Mg}_{1.8}\text{Fe}_{0.2}\text{SiO}_4$, and, in fact, I did. Definitely, I should not rely on references and make this more explicit, which is done in the revised text.

Generally, all equations have analytical form, and properties of any material can be used. Since many parameters are not strictly constrained (A , h , Q_r , and $\dot{\epsilon}(T_0)$), they are varied in the paper.

For plastic flow and heating, I used properties of $\text{Mg}_{1.8}\text{Fe}_{0.2}\text{SiO}_4$, in particular from San Carlos, which is now mentioned in the section title and other places.

Kinetic parameter A was chosen for phase transformation in Zr, because this is the only transformation for which A was found. I added in the text:

"While we study the effect of A on the transformation kinetics (Fig. 4), it is shown below that for large shear strains, the term with A is negligible in the expression for TRIP."

Indeed, it is written in the paper:

"Note that for very large TRIP shear the term $-\sqrt{3}/A$ in Eq. (6)₂ is negligible (Fig. 4(a)), i.e., TRIP shear is independent of any kinetic properties (specifically, parameter A) of strain-induced PT."

For the main parameter ε_o in the equation for TRIP shear, I added in the supplementary material:

”... for Mg_2SiO_4 olivine \rightarrow γ -spinel PT (volumetric transformation strain for complete PT $\varepsilon_o = -0.096$) and for olivine \rightarrow β -spinel PT ($\varepsilon_o = -0.06$) [3,51]. Note that since for Fe_2SiO_4 olivine \rightarrow γ -spinel PT $\varepsilon_o = -0.094$ [51], i.e., practically the same, these numbers can be used for $(\text{Mg}_x \text{Fe}_{1-x})_2\text{SiO}_4$ for any x . For germanium olivine \rightarrow γ -spinel PT, $\varepsilon_o = -0.077$ for Mg_2GeO_4 and $\varepsilon_o = -0.083$ for Fe_2GeO_4 [3,51]. Thus, plots in Fig. 4(a) in the main text for shear strain versus τ/τ_y between curves olivine \rightarrow γ -spinel and olivine \rightarrow β -spinel include results for all these PTs.”

For laboratory experiments, there are no data on shear bands in Mg_2SiO_4 and $\text{Mg}_{1.8}\text{Fe}_{0.2}\text{SiO}_4$. That is why researchers perform experiments with the structural analogs Mg_2GeO_4 and Fe_2SiO_4 (added in the revision due to the Reviewer’s comment), which transform at much lower pressure. I analyzed in detail Mg_2GeO_4 because it has more data and included an estimate of γ for Fe_2SiO_4 (all that could be done based on [15]). Since this γ (and ε_o) is close to that for Mg_2GeO_4 , results should be quite similar. However, even this γ is questionable because the author of [15], Tim Officer, wrote to me that thickness might be overestimated due to probable lateral growth of the transformation zone. He is postprocessing now the results of new experiments, and I will make theoretical estimates for his group.

I also added after estimating γ :

”While faults in Mg_2SiO_4 and $\text{Mg}_{1.8}\text{Fe}_{0.2}\text{SiO}_4$ have not been observed yet, due to the close magnitude of the transformation strain for all $(\text{Mg}_x \text{Fe}_{1-x})_2\text{SiO}_4$ and $(\text{Mg}_x \text{Fe}_{1-x})_2\text{GeO}_4$ for any x (see supplementary materials), similar γ is expected.”

Reviewer’s comment:

2. Some results throughout the paper that lead to other important results are based on what seems to be arbitrary assumptions. For instance, on p. 4 it is assumed that ”... at a higher strain rate increase in stress is slightly higher [than 4.7], 5.8 ...” (by the way, this sentence contains a typo: ”in” instead of ”is”). Why 5.8? How would choosing a different value (high or lower than 5.8) influence the conclusions of this work?

Author’s response:

Since the goal is to receive an increase in strain rate by 18 orders of magnitude, an increase by a factor of $5.8/4.7=1.2$ definitely does not alter any conclusion. This change was used to operate with 10^k without an insignificant pre-factor $1/1.2=0.8$. In the current version, I found a way to eliminate this confusing part without changing the final numbers.

Reviewer’s comment:

3. Similarly, in Supplementary Information (SI), while discussing the solid-solid phase boundary under nonzero strain, the author assumes that a transition from olivine to gamma-spinel occurs below 1695 K, and that from olivine to beta-spinel above 1295 K. The author should comment on how good these assumptions are, and how much below/above the transition temperatures may be. Phase diagrams under nonzero strain are not studied well enough, e.g., how different they are from (quasi-)hydrostatic phase diagrams, so this point may be of some concern to many potential readers.

Author’s response:

I added detailed estimates of the effect of nonhydrostatic stresses on reducing phase transformation pressure below the phase equilibrium pressure. For olivine - β -spinel, the reduction of phase equilibrium pressure does not exceed 0.3 GPa or 2.3%. For olivine - γ -spinel, the reduction of phase equilibrium pressure does not exceed 0.123 GPa or 0.8%. The main effect is plastic strain rather than nonhydrostatic stresses.

In addition, I completely rewrote this section, making it more clear. The key point was to show that system after completing the olivine-spinel PT is in the region of stability of β -spinel; specific numbers are not important. Experimental phase diagrams for Mg_2SiO_4 and $\text{Mg}_{1.8}\text{Fe}_{0.2}\text{SiO}_4$ are added as well. These changes do not affect any final result and conclusion.

Reviewer’s comment:

4. How reliable is the assumption ”s=1” in Eq. (S.1) which leads to the analytic solution $c=1-\exp(-Ae)$, (S.2), for the volume fraction of the high-P phase?

Author’s response:

I added the following justification before Eq. (S.2):

”The only existing experimental results for strain-induced PT kinetics have been obtained for $\alpha - \omega$ PT in Zr [31] and for hexagonal to wurtzitic PT in BN (our yet unpublished data), which were well described at $s = 1$. We will use $s = 1$ here as well, while different s do not change the order of magnitude of the obtained in the main text kinetic estimates.”

As I mentioned in the paper and above, for large shears, kinetic information does not contribute to TRIP shear and following equations.

Reviewer’s comment:

The author operates in terms of strain-controlled kinetic equations. I need to see how they couple to time evolution equations, e.g., in classical nucleation theory, which give $c=1-\exp(-Kt^{**n})$ for the time evolution of c , where t is time and the rate function K strongly depends on temperature. Coupling both equations may provide another insight on a mechanism for a drastic increase in strain rate and temperature that the author suggests in this work.

Author’s response:

It was written and elaborated (in blue) in the paper:

”Thus, the next plastic strain increment leading to new defects and new nuclei at their tips is required to continue PT. That is why (and because of barrierless nucleation, which does not require thermal fluctuations) time is not a governing parameter in a kinetic equation, and plastic strain plays a role of a time-like parameter [16-19,25] (see Eq.(4)).”

There are no any experimental data for any material that demonstrate the time effect on the strain-induced phase transformations. Only for a few dislocations in a pileup, theory [17,18] includes time effect and applies classical thermally activated nucleation theory. In this case, there is no strong acceleration of phase transformations due to plastic deformation, and it is not of interest to the current paper. If additional time-dependend growth does exist, it will further accelerate phase transformation and TRIP, which would make our mechanism more plausible. But, again, it is not observed in experiments, and I do not need it in theory.

Reviewer’s comment:

5. How may the account of realistic crystal structure (polycrystallinity, heterogeneities like grain boundaries, grain edges, crystal defects, etc.) potentially influence the main findings of this work?

Author’s response:

First, I included information about critical stresses for the actual slip systems of olivine to justify orientational softening. Second, strain-induced phase transformations require dislocation pileups against grain boundaries, i.e., polycrystalline structure is required. This was used in our previous works to derive kinetic Eq. (4). I also wrote that ”We do not consider strain-induced reverse spinel→olivine PT because resultant nanograin spinel deforms dominantly by grain-boundary sliding, which does not produce stress concentrators inside the grains.”

Thus, some necessary structural information is effectively included in the continuum models.

Reviewer’s comment:

6. Can the deep-focus earthquake mechanism discussed in this paper be coupled to real subduction dynamics to shed more light on some phenomena that remain not fully understood? E.g., relative orientation of the compression (P-) and tension (T-) axes of non-shallow earthquakes in the upper and lower parts of the downgoing slab, or shear failure in the surrounding rocks as a result of implosion related to the olivine-spinel transition, and some related phenomena like the absence of monopolar rarefaction, etc.?

Author’s response:

The advantage of the suggested model is that it is fully analytical. It can be included as new constitutive equations in any code for studying real subduction dynamics to analyze various existing problems. As a warning, the transformation faulting in my model occurs at the 10 s time scale, while the subduction time scale is million years. Thus, a special multiscale numerical procedure should be developed. However, this is not the goal of the current paper.

Reviewer’s comment:

7. There is a number of typos throughout the paper, e.g., "0.02s (or 20s)" on p. 6; I assume the author means 20 ms. I won’t list them now. If the author resubmits his work and they (some of them) remain in the revised version which I will get a chance to review, then I will get back to this point.

If this is going to be a Nature publication, everything it presents should be very clear to read and understand, and all the assumptions must be justified.

Author’s response:

These are not typos; these are two alternatives. I changed to

"To reach $c = 0.99$, plastic strain $\varepsilon = 4.6/A = 0.2$, which at strain rate 10 s^{-1} (or, alternatively, at 10^{-4} s^{-1}) takes just 0.02 s (or, alternatively, 20 s), instead of millions years without plastic strain."

But I did double-check the grammar and corrected a number of typos.

Reviewer’s comment:

Finally, the paper would gain more scientific weight if the theoretical work it presents were supplemented by possible experimental study on plasticity in Mg_2SiO_4 under P-T conditions close to those in the deep Earth, or some relevant computer simulations. The author analyzes some experimental data on Mg_2GeO_4 , but again, I believe the actual material discussed in Mg_2SiO_4 or $(\text{Mg-Fe})_2\text{SiO}_4$ solid solution.

Author’s response:

As I cleared above and in the revised text, I do use all available published results obtained via experiments or computer simulations on $\text{Mg}_{1.8}\text{Fe}_{0.2}\text{SiO}_4$ and Mg_2SiO_4 . Even if some data are not a well constraint (the same like in all other geophysical papers), I showed that they either do not enter the final equations or varied them to demonstrate that the order of magnitude of the obtained results is not affected.

But generally, I fully agree with the Reviewer and outlined future work at the end of the paper. I submitted a proposal with a clear plan for coupled experimental, theoretical, and computational work on $\text{Mg}_{1.8}\text{Fe}_{0.2}\text{SiO}_4$ San Carlos olivine, without and with some additional minerals, in strain rate range from $10^{-5} - 10^3/\text{s}$. We are expecting a new dynamic rotational diamond cell (dRDAC), which allows covering high strain rates, usually not used in geophysical experiments but required for the model refinement for the processes in a shear band. Since stress, plastic strain, and volume fraction of phases fields in dRDAC are very heterogeneous, we will use our coupled theoretical and computational approaches to extract constitutive equations and material parameters, as we did previously for other materials. This is a three-year plan. Since all previous experimental works on transformation faulting are performed not on $\text{Mg}_{1.8}\text{Fe}_{0.2}\text{SiO}_4$, but on its structural analogs Mg_2GeO_4 and Fe_2SiO_4 , it is clear that this will be high-risk breakthrough experiments and separate papers.

I greatly appreciate the Reviewer’s time and efforts in working on my paper, numerous useful critical comments, and a positive attitude. I took the Reviewer’s comments very seriously and made corresponding changes to the text, which improved the quality of the paper. I also included some changes due to critical comments from the reviewers on my proposal and from comments at geophysical seminars and conferences. Since the Reviewer did not challenge my main conclusions and appreciated the importance of the problem and made progress, I hope that the Reviewer can accept/tolerate my assumptions, responses, and changes.

REVIEWER COMMENTS

Reviewer #1 (Remarks to the Author):

Nothing

Reviewer #2 (Remarks to the Author):

Manuscript#: NCOMMS-22-04706A

Title: Resolving puzzles of the phase-transformation-based mechanism of the deep-focus earthquake

Corresponding Author: Valery I. Levitas

I have read very carefully the author's responses to two reviewer's reports, and the revised version of the manuscript. I'm fully satisfied with the way the author has chosen to respond as well as with the revised paper. Of course, the manuscript could have been improved even further, but I cannot demand anyone to do things beyond what is reasonable. Let us live some room for the researchers that would follow the suit and go one step farther. I'm convinced this is an excellent piece of research that will very likely be of great interest to a wide readership of Nature Communications and will motivate further research in the field of deep earthquakes in particular and phase-transformation-related plasticity in general. I do recommend the publication of this work in Nature Communications.